# Predicting unsafe behaviour from the objective assessment of fatigue manifestation among scaffolders: Evidence from a Quasi-experimental simulation study

Pei Pei Heng[1,2]*, Hanizah Mohd Yusoff[2], Mohamad Ridza Hj Illias[3], Yap Jun Fai[1], Muhammad Fadhli Mohd Yusoff[1], Norizzati binti Amsah[1], Rozita Hod[2]

**1** Institute for Public Health, National Institutes of Health, Ministry of Health Malaysia, Bandar Setia Alam, Selangor, Malaysia, **2** Department of Public Health Medicine, Faculty of Medicine, National University of Malaysia, Bandar Tun Razak, Wilayah Persekutuan Kuala Lumpur, Malaysia, **3** MKRD Training Institute, Malaysian Scaffold Academy, Bangi Hub Suntrack Industrial Park, Bandar Baru Bangi, Selangor, Malaysia

\* hengpeipei@moh.gov.my

## Abstract

### Background

The burden of scaffolding-related accidents was growing. Individual attributes like unsafe behaviour and fatigue have been identified as the important precursors. Understanding the relationship between fatigue manifestation and unsafe behaviour is crucial for developing effective accident prevention protocols. This study aimed to determine the relationship between unsafe behaviour and manifestation of physical and cognitive fatigue during scaffold erection task among scaffolders.

### Methods

A total of 86 scaffolders were recruited into a quasi-experiment with one group pre-post design. Fatigue manifestation was measured objectively at pre-and post-exposure to scaffolding task, employing an assessment protocol consisted of 7 tests. Two fatigue dimensions were evaluated: Physical fatigue (musculoskeletal capacity by hand grip strength, prone plank, trunk flexor endurance and trunk lateral endurance test; postural stability by one leg standing test; joint flexibility by sit and reach test) and cognitive fatigue (simple reaction time test). Throughout the simulation scaffold erection task, unsafe behaviour was rated according to the non-compliance to standard safety protocol. Other independent variables of sociodemographic factors and Pittsburgh Sleep Quality Index score were also included.

### Results

Correlation analysis was significant ($p < 0.001$) for unsafe behaviour with dominant-hand grip strength ($r = -0.57$), one leg standing duration with eyes closed ($r = -0.69$),

**Data availability statement:** Data cannot be shared publicly because of ethical (participants confidentiality) restriction. Even though the data have been anonymized, in combination they are potentially re-identifiable. Researchers of this study have to comply with the requirements of the Universiti Kebangsaan Malaysia Research Ethics Committee. Therefore, data are available upon request to the Universiti Kebangsaan Malaysia Research Ethics Committee (contact via sepukm@ukm.edu.my) for researchers who meet the criteria to access the confidential data.

**Funding:** This research was funded by the Universiti Kebangsaan Malaysia to H.M.Y JEP2022-604.

**Competing interests:** The authors have declared that no competing interests exist.

prone plank duration (r = −0.56), trunk flexor endurance duration (r = −0.61) and reaction time (r = −0.47). Multiple linear regression analysis confirmed the significant predictors of rate of unsafe behaviour including reaction time, one leg standing duration when eye closed, trunk flexor endurance duration and Pittsburgh Sleep Quality Index score.

## Conclusion

These findings will facilitate the development of scaffolding safety protocol for accident prevention, assist industry managers and regulatory decision-makers to govern workers' safe behaviour via the fatigue mitigation approach.

## Introduction

Globally, the building and construction industry is recognized as a remarkably hazardous industry, which documented a significant number of fatality following occupational accidents [1]. About 80% of the construction mishaps were contributed by individual error such as the unsafe behaviour [2]. The Malaysian occupational accident statistic documented a total of 45 construction deaths and 8 cases of permanent disability throughout year 2023 (DOSH Malaysia 2023). When most construction accidents worldwide reported related to collapsed or malfunctioned scaffold [3], local scaffolding-related accidents have also been widely reported [4,5]. A majority of these accidents were primarily due to the root cause of human unsafe act like poor inspection, improper assembly of scaffolding materials, poor safety culture and behaviours [3].

Scaffolds aretemporary structure extensively used to support building works at heights and places with poor access. Scaffold safety was not being given sufficient attention due to its impermanent usage at the construction site [6]. Scaffolders are the main workforce who handle the scaffolding task like erection and dismantling. Therefore, either deviation from the standard safety protocol or safe behavioural working performance among scaffolders will leave an enormous impact on the construction of unsafe scaffold. Two ways of scaffolding-related mishaps could result from the faulty scaffold. Firstly, fall from height, slip and trip incident and falling objects among the scaffolders following their poor safety performance during scaffolding task. Moreover, the unsafe scaffold might have the risk of collapse, injuring both the public and other scaffold users on building sites.

The scaffolding sector poses a greater risk to worker safety, because of the involvement of manual handling labourers with physically demanding tasks that prone to work fatigue [7]. Fatigue is a condition where individual manifested with a diminished capability to perform activities at the desired level, because of physical or cognitive exhaustion, or both [8]. Other than physical weariness, scaffolding task might result in cognitive fatigue, as the works require an adequate level of alertness during the sequential procedural steps. Collectively, the reduced physical capabilities and lapsed in cognitive function following fatigue may compromise task performance [9]. In the occupational setting, the rate of human error is associated with the level of

weariness. Workers might be unfit for duty execution while fatigued; leaving an impact on safety performance which raises the likelihood of ill-health and injury [10]. Individual factor like mishandling due to safety violations often being documented as an causative factor of scaffolding accidents [4]. The existing accidents' prevention strategies focused on the passive safety counter measures like Personal Fall Arrest Systems in accordance with respective safety regulations [11]. Nonetheless, this resolution did not significantly improve the scaffolding safety statistics. There is still a dearth of research on individual characteristics, particularly the unsafe behavior [12].

Given the growing burden of the scaffolding-related accidents, the gap of knowledge on the relationship between fatigue manifestation and unsafe behavior need to be ascertained. This knowledge is important in the development of fall prevention protocol and targeted fatigue intervention among scaffolders. Additionally, fatigue could be an individual attribute of occupational accident; while the unsafe behavior had been also recognized as one of the most prominent factors of scaffolding-related accidents [13]. Up to date there is no confirmation on the relationship between fatigue and safety behavior from the study among scaffolders. Furthermore, due to multidimensionality of fatigue (Phillips 2014), self-reported fatigue is prone to reporting bias hence does not reflect the true fatigue level. The employment of objective assessment scale for fatigue manifestation is therefore more reliable in detecting, and subsequently to effectively manage fatigue. The potential on-site assessment of fatigue among scaffolders should be identified. This assessment must fulfil specific criteria including ease to administer on scaffolding site, rapid, not invasive, unlikely to interrupt workflow and tailored to the specific work scenario [9,14]. With the refinement of relationship between fatigue and unsafe work performance, the application of fatigue assessment can guide the Site Safety Supervisor to ensure fitness for daily duty, as well as to predict the unsafe behaviour before scaffolders are allowed to handle risky tasks. This study aimed to determine the relationship between the physical and cognitive manifestation of fatigue, which are assessed objectively, and the unsafe behavior during a scaffold erection task among the scaffolders.

## Materials and methods

This is a quasi-experimental study with one-group pre post design, conducted from 18 December 2023 until 30 April 2024. All scaffolders from various scaffolding company of the *Lembah Klang* region, Malaysia, who attended scaffolding training at the certified scaffolding training site (MKRS Training Intitute, *Bandar Baru Bangi*), throughout the study period were invited to participate.

### Sampling

The consecutive non-probability sampling was applied as the most practical technique. Sample size was estimated by two-correlated proportions formula [15], taken into account both physical and cognitive fatigue [9]. Prior data reported 0.34 (P1) as the proportion of fatigued construction workers who are more likely to cause manifestation of fatigue symptoms and work errors; while P0 as 0.11 [9]. To reject null hypothesis with a power of study (0.8), probability of Type I error (0.05), anticipated dropout rate (25%), a total of 86 sample was recruited. The inclusion criteria were: (i) adults aged 18–60 years, (ii) Malaysian and non-Malaysian, (iii) able to understand *Bahasa Melayu* or English; (iv) No communication impairment in terms of language barrier. They were excluded if fulfilling any exclusion criteria: (i) who had been clinically diagnosed with sleep disorder (insomnia, sleep apnoea, narcolepsy) or (ii) pathological fatigue secondary to comorbidity.

### Study protocol

The Patient Information Sheet and written consent were distributed prior to study. During the training module, there was no hands-on exposure to scaffolding task from day 1 to day 4. From day 1 to day3, only theory classes were conducted to deliver relevant learning module of the fundamental scaffolding knowledge on scaffold components and fittings. Participants were taught on the safety sequential steps associated with erection and dismantling of the basic static scaffold tower with 2-lifts. On day 4, participants were given introductory step-by-step live demonstration by the trainer on site.

Practical session was initiated on day 5 which indicated an exposure phase, where participants were exposed to the repeated hands-on cycle of scaffold erection-dismantling procedures from 9.00 am to 6.00 pm, with 3 standardized breaks in between. Hence, this exposure on day 5 was hypothesized to generate fatigue. A 14-hours of overnight rest were allowed until they came back to training site the next day (day 6). Fatigue generation during work exposure on day 5, and the carry-over effects of fatigue from one work day (day 5) to the next before scaffolders are allowed to initiate next task (day 6), have been affirmed. Our evidence on pre-post deterioration in fatigue parameters during this study phase had been published earlier by Pei et al. 2025 [16]. In the post-exposure state where fatigue had developed (day 6), fatigue manifestation was measured early in the morning, followed by their participation in simulated scaffold erection task from 9.00am-12.00 pm.The impaired safety performance in term of unsafe behavior was observed and evaluated at individual level throughout this simulated work (Fig 1). This task was conducted in groups, which mimics the real scaffolding setting, in which 4 participants were grouped in one group to erect one scaffold tower. Each group was evaluated by one evaluator from the research team who have received standardized training from the scaffolding training academy on the criteria of unsafe behaviour detection and assessment. For the assessment of fatigue manifestation, the same evaluator performed the similar test for all individual with standardized measurement method in order to minimize the measurement variation. During the quasi-experiment, the scaffolding variables (type of scaffold, height, working time, working surface, PPE provision, fall arrest measure) and the environmental factors (wind, rain, heat) that may contribute to the impaired safety performance, were controlled on training site. After the simulation protocol, participants were provided with isotonic drinks and energy bar to recuperate.

The safety of the respondents was thoroughly monitored by the trainer of the certified training centre. All the fall arrest safety features applied by the training academy were secured. The task was instructed to pause immediately by safety trainer upon detection of any incompliance with the safety measure. Participants were given reminder to adhere to the appropriate safety measures before the simulated task was allowed to go on. At the same time, unsafe behaviour was being rated.

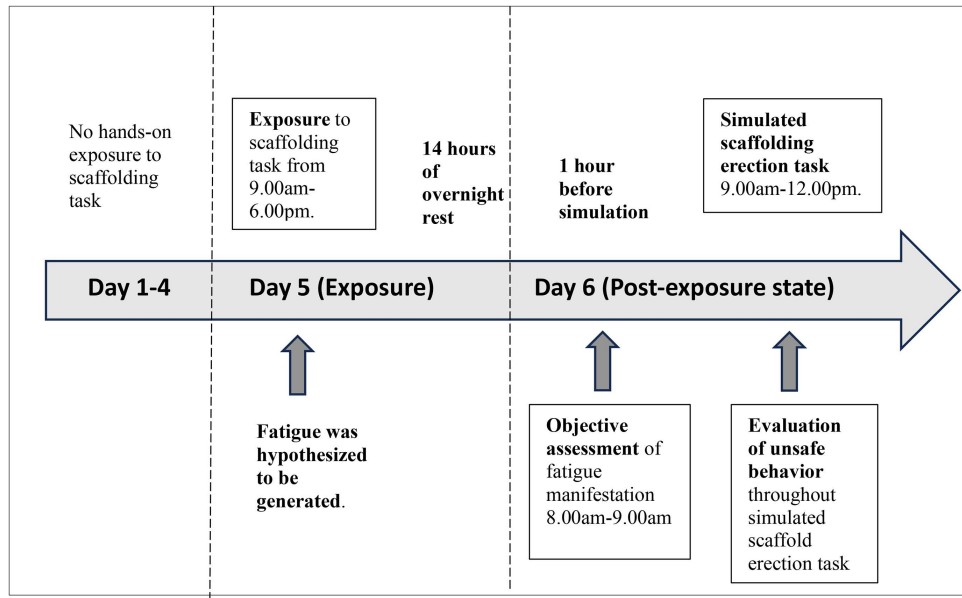

**Fig 1. Study protocol illustration.**

## Study instruments and measures

The dependent outcome was measured as the rate of deviated safe behaviour, (n/38) x100%, whereby (n) was total number of non-compliance based on the 38-standard protocols in the validated procedural checklist for unsafe behavioural evaluation during basic static scaffold tower erection, which was developed in an earlier phase of the study. The complete checklist is available in the supplementary S1 Table. The details of checklist development via Focal Group Discussions and validation via Fuzzy Delphi Method, was published earlier [17]. Our procedural checklist demonstrated high experts' consensus for each construct with threshold value (d) ≤ 0.2; and for all items with agreement of ≥75%. The average fuzzy numbers documented ranging from 0.588 to 0.8. None of the items with the lowest ranking was discarded as all items perfectly fulfilled all prerequisite and obtained excellent experts' agreement.

The independent variables of fatigue manifestation were measured objectively, covering physical fatigue (musculoskeletal capacity, postural stability, joint flexibility) cognitive fatigue (reaction time). These elements had been documented deteriorated during state of fatigue in construction field. The test protocol consisted of a total of 7 tests: hand dynamometer for hand grip strength [18]; prone plank test for core strength and stability: trunk flexor endurance test and trunk lateral flexor endurance (side bridge) test for trunk and back endurance [19]; sit and reach test for lower limb joint flexibility [20]; one leg standing test with eyes opened and eyes closed for postural stability [21]; and visual red light-green light simple reaction time test [22]. The evidence on the deterioration of fatigue parameters post-exposure which were being measured objectively was published in an earlier phase of study [23].

The test order adhered to a justified sequence in order to assure the validity of each test element. The test protocol was initiated with cognitive fatigue measure, because physical activities have been evidently reported to facilitate mental fatigue [24]. Among the physical fatigue measures, postural stability was assessed before the musculoskeletal capacity and joint flexibility because postural sway might occur as a result of fatiguing protocol to specific muscle groups [25]. Furthermore, muscular endurance tests were performed after the strength tests. Endurance tests are higher in intensity that cause muscle soreness, and consume longer duration [26]. Verbal instruction demonstration was delivered by a trained research team member to all participants before commencing the test. The sequence of test protocol is summarized in Table 1. The measurement error was controlled by the use of valid test protocol. The equipment like hand dynamometer was well-calibrated. The same gadget was used for the on-screen cognitive test for all participants.

Apart from that, the questionnaire was distributed to collect data on sociodemographic and job characteristics including age, gender, ethnicity, educational level, comorbidities, smoking status, alcohol consumption, work experience and previous formal scaffolding training. The validated Malay Version of the Pittsburgh Sleep Quality Index (PSQI-M) had also been distributed to obtain input on sleep hygiene [30].The PSQI consisted of 7 domains (sleep quality, latency, duration, efficiency, disturbance, medication and daytime dysfunction), with total score range from 0 to 21. A score >5 is considered as poor sleeper, while score of 1–5 indicates good sleeper [31].

## Data management and analysis

The statistical analysis was performed with SPSS Version 22. Results were presented with 95% confidence interval (CI), with p-values of 0.05 for the statistical significance. Data was screened and cleaned. Descriptively, the individual characteristics were illustrated in frequency (n) and percentage (%).

The correlation between objective measurement of fatigue manifestation and the rate of unsafe behavior tested with Spearman correlation test, in view of rate of unsafe behavior was not normally distributed. The correlation coefficient, (r) was used to describe the degree of linear relationship. The r of 0–0.19 is regarded as very weak, 0.2–0.39 as weak, 0.40–0.59 as moderate, 0.6–0.79 as strong and 0.8–1 as very strong correlation. In the simple linear regression analysis, the p-value of <0.05 in ANOVA table indicated the fit model. The linear regression model was valid as all assumptions that checked on residuals were fulfilled. Multiple Linear Regression only considered independent variables with significant results in the simple linear regression (p < 0.05). The confounders investigated including age and other categorical

**Table 1. Test sequence for the objective assessment of fatigue manifestation.**

| Test sequence | Fatigue dimension | Assessment method/ Test | Test protocol | Measurement |
|---|---|---|---|---|
| 1 | Cognitive (Reaction time) | Visual Red light green light simple reaction time test | Subject was required to stop using screen device 15 minutes prior to the test. The 14-inches wide screen laptop was used and subject responded by clicking mouse with dominant hand. First, subject clicked on the start button to begin. When the stoplight turned from red to green, subject clicked on the button as quick as possible. Reaction time was recorded for a total of 5 attempts and the average time of reaction was displayed [22]. | millisecond |
| 2 | Physical (Postural stability and control) | One leg standing test with eyes opened and closed | Subject was standing upright with arms lowered alongside with hips. Firstly, subject lifted up any one leg with eyes opened, and next the same test with eyes closed. The countdown stopped when the lifted leg touched the floor or when subject moved arms away from body to stabilize the position [27]. | second |
| 3 | Physical (Joint flexibility) | Sit and Reach (SR) Test | The subject sat on the floor their back, hips and knees straight. With the legs together, both soles of the feet were positioned flat against a box. Subject extended arms with palms down and lightly touched the index fingers together. Subject was asked to bend forward to reach as far forward as possible while keeping the knees extended. The distance was measured between the fingertips and the point at which the feet contacted to the box. 0 cm represented the position of the feet against the box, with larger values for higher flexibility [28]. | centimetre |
| 4 | Physical (Hand grip strength) | Hand dynamometer test for dominant and non-dominant hand | Hand dynamometer was calibrated prior to study. Subject holds dynamometer in one hand, standing upright with elbow flexed at 90-degree angle. The equipment's grip was adjusted accordingly to ensure subject exerted force by only last four phalanges to the handle. The maximal strength was performed for 5 seconds, with 3 attempts on both hands. The highest strength was recorded. Rest period of 30 seconds was allowed between the attempts [29]. | Kilograms |
| 5 | Physical (Core strength and stability) | Prone plank test | Subject was positioned prone with elbow positioned at a 90-degree angle. Subject then raised pelvis from the floor and maintain a flat position, the test was terminated when participant unable to hold the position [19]. | second |
| 6 | Physical (Trunk flexor endurance) | Trunk flexor endurance test | Subject sat in a semi-reclining position with hips and knees at 90-degree. Both arms were placed across the chest. Subject lean beside a board that is kept in an incline 60-degree angle, whereby the head was maintained in a neutral position. After the board was removed, the position must be maintained using the abdominal muscles to sustain a flat-to-neutral spine, without arching the back. Any evident changes in the position of the trunk such as rising in the low-back arch or an aberration from the neutral position terminated the test [19] | second |
| 7 | Physical (Trunk lateral flexor endurance) | Trunk lateral flexor endurance test (side bridge test) | Subject was on one side of the body, both legs extended and the feet in front of another. The elbow of the supporting arm (the arm which is on the lower side during side-lying) was placed below the shoulder with the forearm facing out. The other upper limb was placed on the chest. Subject was instructed to raise the hip. The trunk was supported only by foot, and the elbow/forearm of the lower arm. Any evident changes in the position of the trunk such as rising in the low-back arch or an aberration from the neutral position terminated the test [19]. | second |

dichotomous variables which were all being coded numerically, coding "1" for higher risk group. For example, smoking status (smoker = 1; non-smoker = 0), alcohol consumption (alcoholic = 1; non-alcoholic = 0), PSQI category (poor sleeper = 1; good sleeper = 0), working experience in scaffolding work (less than 5 years = 1; at least 5 years = 0). No interaction and no intercorrelations were detected between all independent variables, with the tolerance value above 0.6 and Variance inflation factors below 10. This indicated a stable regression model. All independent predictors were selected and run with three modes: enter, forward, backward and stepwise method. The biggest model with all significant selected variables was chosen as the preliminary model. All residual-assumptions of linearity, independence, normality and equal variance and absence of outliers were met before the final model was accepted.

### Ethical clearance

The ethical approval was obtained from the Research Ethics Committee of University Kembangan Malaysia (JEP-2022–604, Project Code: FF-2023–298). Written consent was obtained from all participants before recruitment. Participation was based on voluntarily basis with the privacy, anonymity and confidentiality of personal information adequately maintained.

### Results

The descriptive illustration of individual characteristics for all participants (n = 86) is presented in Table 2.

The correlation between all tests for the objective assessment of fatigue manifestation and the rate of unsafe behaviour is tabulated in Table 3.

There was a weak negative correlation (r = −0.26, p = 0.02) between non-dominant hand grip strength and rate of unsafe behaviour. For dominant hand, the correlation was moderately strong negative (r = −0.57, p < 0.001). There was no significant correlation between one leg standing duration when eye opened, with the rate of unsafe behavior (p > 0.05). Nevertheless, when the test was repeated with eyes closed, there was a strong negative correlation (r = −0.69, p < 0.001). A reduced duration of one leg standing time with closing eyes was correlated with higher rate of unsafe behaviour. The prone plank duration recorded a moderately strong negative correlation(r = −0.56, p < 0.001) with rate of unsafe behaviour; trunk flexor endurance duration was strongly and negatively correlated to rate of unsafe behavior (r = −0.61, p < 0.001). However, no significant correlation was reported for two tests: side bridge test and sit and reach test. Cognitively performance measure revealed a moderately strong positive correlation (r = 0.47, p < 0.001) between simple reaction time and rate of unsafe behaviour. A longer time of reaction was correlated with higher rate of unsafe behaviour during simulated scaffolding erection work.

Table 4 presents the quantification of the relationship between all independent variables and rate of unsafe behaviour during simulated scaffold erection task. Simple linear regression showed significant relationship between rate of unsafe behaviour and reaction time, non-dominant hand grip strength, dominant-hand grip strength, one leg standing duration with eye closed, prone plank duration, trunk flexor endurance duration, work experience and PSQI score.

Multiple Linear Regression had generated quantitative prediction of unsafe behavior. When all counfounders were adjuested, there was significant linear relation between reaction time, one leg standing duration with eye closed, trunk flexor endurance duration and PSQI score with the rate of unsafe behaviour (Table 5).

Rate of unsafe behaviour (%) = −0.83 + (0.03*reaction time) -(0.30*one leg standing duration when eye closed) -(0.04*trunk flexor endurance duration) + (1.18*PSQI score).

Scaffolders with every increase of 100 milliseconds in reaction time, the rate of unsafe behaviour increased by 3% (95% CI 1.4,4.6, p < 0.001). Those having 10 second less in the one leg standing duration when eye closed, the rate of unsafe behaviour increased by 3% (95% CI −4.34, −1.58, p < 0.001). Those with 10 second less in the trunk flexor endurance duration will increase the rate of unsafe behaviour by 0.4% (95% CI −0.72, −0.08, p < 0.05). Other than that, the poor sleeper with PSQI score >5 had significant 1.18% increase in unsafe behavioural rate than the good sleepers with PSQI score of 5 and below (95% CI 0.72, −1.40, p < 0.05).

### Discussion

#### Grip strength of dominant hand

The correlation between rate of unsafe behaviour and dominant-hand grip strength was significantly stronger than non-dominant hand strength. This indicated that functional performance of upper extremity is influenced by hand dominance [32]. A number of specific safety performances in scaffold erection task, for example fixing, screwing or tightening of materials, require only motor activity of single hand, typically the dominant-hand, resulting in a more remarkable functional performance impairment [33].The massive upper limb tasks causing a more prominent muscular fatigue of dominant-hand

**Table 2. Demographic characteristics of participants (n = 86).**

| Demographic variables | n (%) | Mean ±SD |
|---|---|---|
| **Age Group (years)** | | 32.2 (±8.68) |
| 19-29 | 44 (51.2) | |
| 30-39 | 27 (31.4) | |
| 40-49 | 10 (11.6) | |
| 50-59 | 5 (5.8) | |
| **Gender** | | |
| Male | 82 (95.3) | |
| Female | 4 (4.7) | |
| **Ethnicity** | | |
| Malay | 49 (57.0) | |
| Chinese | 3 (3.5) | |
| Indian | 34 (39.5) | |
| **Educational level** | | |
| Secondary | 9 (10.5) | |
| Tertiary | 77 (89.5) | |
| **Comorbidity** | | |
| Yes | 3 (3.5) | |
| No | 83 (96.5) | |
| **Smoking status** | | |
| Never | 21 (24.4) | |
| Quit | 26 (30.2) | |
| Active smoker | 39 (45.3) | |
| **Alcohol consumption in past 30 days** | | |
| Yes | 18 (20.9) | |
| No | 68 (79.1) | |
| **Job title** | | |
| Site supervisor | 37 (43.0) | |
| Scaffolding operators/ Scaffolders | 38 (44.2) | |
| Others (technical maintenance and quality team) | 11 (12.8) | |
| **Experience in scaffolding work** | | |
| Less than 5 years | 57 (66.3) | |
| At least 5 years | 29 (33.7) | |
| **Previous training** | | |
| Yes | 22 (25.6) | |
| No | 64 (74.4) | |
| **Working hours per day** | | 8.3 (±0.98) |
| **Number of work breaks** | | 2.7 (±0.48) |
| **Total break duration (minute)** | | 86.3 (±12.16) |
| **Involvement in past work incident** | | |
| Yes | 5 (5.8) | |
| No | 81 (94.2) | |
| **PSQI Global Score** | | |
| Good sleeper (score 1–5) | 61 (70.9) | |
| Poor sleeper (score>5) | 25 (29.1) | |

**Table 3. Correlation between all tests of objective assessment for fatigue manifestation and rate of unsafe behavior (n = 86).**

| Objective assessment for fatigue manifestation | | Rate of unsafe behavior (%) |
|---|---|---|
| Hand grip strength (non-dominant) | Correlation Coefficient (r) Sig. (2-tailed) | −0.262 0.015* |
| Hand grip strength (dominant) | Correlation Coefficient (r) Sig. (2-tailed) | −0.571 <0.001* |
| One leg standing duration with eye opened | Correlation Coefficient (r) Sig. (2-tailed) | −0.053 0.63 |
| One leg standing duration with eye closed | Correlation Coefficient (r) Sig. (2-tailed) | −0.685 <0.001* |
| Prone plank duration | Correlation Coefficient (r) Sig. (2-tailed) | −0.563 <0.001* |
| Trunk flexor endurance duration | Correlation Coefficient (r) Sig. (2-tailed) | −0.609 <0.001* |
| Trunk lateral endurance duration (side bridge test) | Correlation Coefficient (r) Sig. (2-tailed) | −0.075 0.49 |
| Sit and reach flexibility distance | Correlation Coefficient (r) Sig. (2-tailed) | −0.127 0.24 |
| Reaction time | Correlation Coefficient (r) Sig. (2-tailed) | 0.469 <0.001* |

Spearman correlation test.

*Correlation is significant at the p < 0.05 (2-tailed).

among scaffolders. The musculoskeletal system will be overloaded by repeated contraction of wrist extensor and finger flexors, during lifting, carrying, gripping and holding an object, collectively posted a negative impact on palmar grip strength [34]. Therefore, it could be concluded in such a way that peripheral muscular fatigue following scaffolding task had led to the reduced hand grip strength, more prominent over the dominant-hand. Given that muscular fatigue in overall decreases concentration and alertness, this resulted in significant negligence or risky behavior at work [35].

### One leg standing test with eye closed

Ample of studies discussed about fatigue-related impairments on balancing performance. We reported a significantly strong correlation between the test duration when eyes closed and unsafe behavioral rate, compared to no correlation when similar test was performed with eye-opened. The postural balancing strategy is vision-dominant. The visual factor plays more important role than any other factors in maintaining a stable posture [36]. Vision can improve bipedal upright stability during standing and locomotion as part of the integrated sensory feedback system [37]. Ocular cues especially the percentage of eye closure had been cited as a strong indicator for fatigue [38]. Other eye metrics like eye blinking, increased blink duration and shorter interval time between blink, have also been reported effective for fatigue detection [39]. The accuracy of these ocular predictors had been affirmed by the electrooculography examination [40]. Given that balancing strategy is vision dominant, thus these eye metrics disturbances while fatigue will compromise the postural control.

One leg standing duration with eyes closed had been identified as a significant predictor of unsafe behavior in the linear regression model. This finding implies this test is more accurate to be conducted with eyes closed in the scaffolding industry in predicting unsafe behavioral performance. Our data had confirmed that static balance might be impacted by visual abnormalities that interfere with the perception of visual cues. Hence, we recommended the importance and relevance of periodic vison examination among the scaffolders.

**Table 4. Quantification of the relationship between all objective assessment for fatigue and rate of unsafe behaviour (n = 86).**

| Variables | Simple Linear Regression (SLR) | | | |
|---|---|---|---|---|
| | b | t | *P* value | 95% CI (Lower, Upper) |
| Constant<br>Reaction time (ms) | −8.880<br>0.046 | 5.150 | <0.001*** | 0.028,0.064 |
| Constant<br>Hand grip (Non-dominant)(kg) | 10.982<br>−0.110 | 3.092 | 0.003** | −0.180,-0.039 |
| Constant<br>Hand grip (Dominant)(kg) | 14.983<br>−0.221 | 6.023 | <0.001*** | −0.295,-0.148 |
| Constant<br>One leg standing test (eye opened)(s) | 7.947<br>−0.036 | 1.022 | 0.31 | −0.107,0.034 |
| Constant<br>One leg standing test (eye closed)(s) | 10.854<br>−0.534 | 9.642 | <0.001*** | −0.644,-0.424 |
| Constant<br>Prone plank (s) | 11.660<br>−0.081 | 6.285 | <0.001*** | −0.107,-0.055 |
| Constant<br>Trunk flexor endurance (s) | 12.634<br>−0.100 | 7.147 | <0.001*** | −0.128,-0.072 |
| Constant<br>Age (year) | 7.120<br>−0.001 | 0.018 | 0.99 | −0.082,0.081 |
| Constant<br>Educational level (below tertiery) | 2.852<br>1.112 | 2.566 | 0.09 | 0.642,5.063 |
| Constant<br>Smoking status (Yes) | −0.520<br>0.707 | 0.734 | 0.46 | −1.926,0.887 |
| Constant<br>Alcohol consumption (Yes) | 0.289<br>0.868 | 0.333 | 0.74 | −2.015,4.137 |
| Constant<br>Work Experience (less than one year) | 6.078<br>1.536 | 2.110 | 0.03* | 0.099, 2.985 |
| Constant<br>PSQI Score (poor sleeper) | 1.862<br>1.751 | 2.479 | 0.02* | 1.369,3.356 |

Dependent variable: rate of unsafe behaviour (%)

b: Crude regresison coefficient

*SLR is significant at p < 0.05

**SLR is significant at p < 0.01

***SLR is significant at p < 0.001

## Prone plank test

A decreased prone plank duration was significantly correlated with unsafe behavioral rate. The association between core strength and postural stability had been widely reported. Core muscle strength play an integral role in anchoring the center of gravity, which provides an ability for balancing and steadiness [41]. The scaffolding tasks involving working at height, prolonged use of body harness and manual lifting, which require a good core strength to maintain an optimum stability throughout work [42]. Previously, biomechanics study reported large moments were created in the trunk during lifting activity. On the other hand, the task height was also proven increased muscular activity of erector spinae which is one of the main muscles of core [43]. Realizing the strong correlation between the diminished core muscle strength and

Table 5. Factors associated with the rate of unsafe behaviour (%) (n = 86).

| Variables | SLR[a] | | | MLR[b] | | |
|---|---|---|---|---|---|---|
| | b[c] | 95%CI | P value | Adj.b[d] | 95%CI | P value |
| Constant | | | | −0.83 | | |
| Reaction time (ms) | 0.046 | 0.028, 0.064 | <0.001 | 0.030 | 0.014, 0.046 | <0.001 |
| One leg standing test (eye closed) (s) | −0.534 | −0.644, −0.424 | <0.001 | −0.296 | −0.434, −0.158 | <0.001 |
| Trunk flexor endurance (s) | −7.147 | −0.128, −0.072 | <0.001 | −0.040 | −0.072, −0.008 | 0.03 |
| PSQI Score | 1.862 | 1.369, 3.356 | 0.02 | 1.176 | 1.017, 1.395 | 0.03 |

[a]Simple Linear Regression

[b]Multiple Linear Regression ($R^2 = 0.718$, The model reasonably fits well. Model assumptions are met. There are no interactions between independent variables. There is no multicollinearity problem).

[c]Crude regression coefficient

the reduction in safe behavior, the integration core fitness training is therefore important to enable workers to learn the optimum way of supporting body weight, navigate challenging terrains, and effectively maneuver equipment while handling at-height work [43].

Nevertheless, this test documented insignificant relationship with unsafe behavior in the multiple linear regression, most likely due to the influence of other factors. We postulated that strength might recover at a more rapid rate compared to endurance. The fatigue recovery varies between individual of various physiological and metabolic status [44]. The change in muscle performance in respond to fatigue protocol indeed is a complex mechanism, consisted of a range of molecular properties such as action potential, extracellular and intracellular ions, and metabolites [45]. Moreover, individual nutritional status plays an important role in sustaining muscle strength. For example, an acute carbohydrate manipulation promises an enhanced strength [46]. It is therefore recommended that future study to look into the mediating effects of metabolic and nutritional factors on the relationship between core strength and rate of unsafe behaviour.

### Trunk flexor endurance test

The trunk flexor endurance was reported as a significant predictor of rate of unsafe behaviour, after adjusted for confounding effects. The endurance test was recognized as a more exclusive goal than strength test in quantifying trunk fatigue [47].Previous study among construction workers highlighted a linear relationship between fatigue levels and error rates. When a worker's fatigue level accumulated and exceeded certain limit, they were more likely to have failure in the perception of hazard [14]. Trunk flexor endurance is affected most by excessive trunk flexion. For example, the lower lifting activity from ground level to higher level [48]. This corresponded to the scaffolding tasks because all scaffold materials like metal tubes, couplers, boards and screws are all being placed on the ground. Scaffolders will have to perform lower lifting each time they want to reach for those materials. Previous study had also reported a significant association between bending/lifting activities that cause excessive trunk flexion and trunk flexor muscle fatigue [49].

### Trunk lateral endurance test (side bridge)

This test was performed as the final test in the protocol; hence the test order effects might have existed. Literature had reported the psychological impact of test order on the outcome. Respondents typically tend to maximize effort in those test that was administered first, compared to the tests being performed later [50].

Moreover, the test protocol was considered generating fatigue as it required upper body strength and interscapular muscle endurance [51] to support and sustain the bridging position. Hence, acknowledging individuals are only supported on one side, the condition of upper extremity could a limiting factor in test performance [52]. Greene et al. documented over half of the athlete participants reported upper extremity pain or tiredness as the reasons for test termination; versus

only minimum respondents who reported trunk side fatigue [53]. These limiting factors must be given serious consideration in the future researches.

### Sit and reach test

The sit and reach test for joint flexibility showed no significant correlation with rate of unsafe behaviour, which indicated the test is not a relevant indicator of physical fatigue measure among scaffolders. The scaffold erection protocol involves many stretching motions, which have definitely improved the joints' range of motion. Although no similar study was conducted in the construction field, previous study on the effect of physical exertion on lower limb flexibility among professional soccer reported an increase of lower limb range of motion 24 hours post exertion [54]. Moreover, physical exertion had been shown to stimulate the stretch reflex activity; at the same time promote a better tolerance to joint stretchability [55].

### Simple reaction time test

Our study reported a moderately strong positive correlation between reaction time and rate of unsafe behavior. Simple reaction time was also a significant predictor for unsafe behavior in the regression model. Reaction time is a measure of cognitive exhaustion. Past study had concluded the association between an increased cognitive fatigue and the tendency to deviate from safety acts and make risky decisions [56]. Our findings supported by a simulation and computational experiment in China which demonstrated that construction workforce with a higher degree of conformity mentality is more inclined to initiate risky behaviours. This was due to the positive association between cognitive process, decision making and the risk perception [57].

In addition, the classic theory of Mood Maintenance Hypothesis had implied that negative emotion compromises focus and information processing thus reduces the estimation of risk events [58]. On the other hand, literature also reported a positive relationship between fatigue and emotions control [59]. It could be concluded that, in the state of fatigue, intense emotions are triggered hence attention allocated to identification of safety hazards will be reduced. As a result, workers tend to have a delayed judgment, with a concomitant decrease in the accuracy of safety hazards recognition and the perception [59].Therefore, other than the screening of reaction time prior to duty, it is also reasonable to regularly assess scaffolders' psychological wellbeing. Supervisors and site inspectors should assess workers' physical and mental readiness at the start of each shift, recognizing that the demanding nature of construction tasks requires full alertness and fitness.

### Individual attributes of unsafe behaviour

The PSQI score was reported as a significant predictor for unsafe behavior. The PSQI tool is a validated global instrument to evaluate an individual's sleeping patterns, the impact of sleep disturbances on daytime functioning, and usage of sleep aids within past a month [31].

Our result was in line with Korean study which revealed that poor sleeper was more prone to sleep for shorter duration before work day, showed poorer sleep latency, with greater daytime dysfunction, more likely to have mental health disturbance especially the depressive symptoms and also a higher physical fatigue level [60]. Poor sleep is a critical issue that construction workers often deal with, because the cumulative weariness is more likely to cause them to sleep for shorter period of time at night or experienced the reduced sleep efficiency [61]. The association between poor sleep hygiene and fatigue in construction industry had been well established. Poor sleepers were associated with an increased risk of fatigue, exhaustion and pain among the construction workers of Saudi Arabia [62] and Hong Kong. In Taiwan, it was highlighted that construction accidents were mainly attributed by poor sleep hygiene [63]. Another Indian study via an integrative interpretive structural modeling analysis, had discovered sleep quality as one of the root causes of unsafe behavior among construction workers [64]. These evidences, collectively recommended that sleep-orientated

intervention in scaffolding industry should be considered with diverse options to ensure an optimum fatigue management and mitigation.

Educational level and work experience did not significantly predict unsafe behaviour in our study. Scaffolding task represents skilfulness and competency. The is the presence of other more important predictors of safety performance and compliance such as on-job training and education process which improvs the safety judgment, job-specific safety knowledge, on-task skills, and motivation of self-learning [65]. In this context, emphasis must be consistently placed on ensuring certified scaffolding procedure, third-party safety inspections, and rigorous enforcement of safety protocols at the organizational level.

Zooming into the lifestyle factors, past study documented smokers easily distracted at work during smoking. Consequently, it had caused violation in the standard operating procedure and led to accidents [66]. On the other hand, alcohol abuse has been identified as an influential factor in unsafe behavior [67]. Nonetheless, different from the previous studies, our participants in present study were not allowed to smoke or drink, and work simultaneously, making the influence of these lifestyle factors could not directly assessed on-site.

### Strength and limitations

This is the first study employed objective measurement tool to assess fatigue manifestation, and refined its relationship with unsafe behavioural rate, specifically among the scaffolders.. The generalizability was maintained to the best of our limit, as there was adequate examination of work characteristics among scaffolding workers. The experimental setting was also mimics the real scaffolding site therefore, it could be possible to interpret findings among other groups of scaffolding workforces. Nevertheless, the inference from our sample to the target population of scaffolders might be slightly inadequate, because present study recruited participants who attended training at training site in order to ease the conduction and control of the quasi-experiment. There might be variations in term of individual sociodemographic and job characteristics like age group and work experience. There was also a lack of thorough control of the climatic factors as strict monitoring of environmental parameters such as humidity, radiant temperature and wind speed were not practical in the field-experiment. Our study somehow was a preliminary exploration on this research topic. Based on all positive outputs generated and realizing all limitations, we therefore recommended future studies to be expanded in the real scaffolding site.

### Conclusion

The quasi-experiment had refined the significant relationship between physical and cognitive fatigue manifestation, and the rate of unsafe behaviour. The prediction model had confirmed four significant fatigue predictors of unsafe behaviour, including simple reaction time, one leg standing duration with eyes closed, trunk flexor endurance duration and PSQI sleep quality screening, indicating these testscan be practically used as the on-site evaluation of fatigue and anticipate safety performance before scaffolding task is assigned to scaffolder. Thestudy findings will assist industry managers and regulatory decision-makers to put in place appropriate measures to govern workers' safe behaviour, besides guiding the formulation of fatigue intervention and training among scaffolders, which eventually helps in accidents' prevention. As an extension to present study, future studies are needed to explore the tests' application and feasibility of conduction in the real scaffolding setting.

### Supporting information

**S1 Table. Validated Checklist for Individual Evaluation of Unsafe Behaviour during the 2-Lifts Basic Static Scaffold Tower Erection Protocol.**
(DOCX)

## Acknowledgments

We thank the Director General, Ministry of Health Malaysia for the publication approval. We would like to thank the Dean of the Faculty of Medicine, Universiti Kebangsaan Malaysia and the Department of Public Health Medicine, Faculty of Medicine, Universiti Kebangsaan. Malaysia for permission to conduct this study. We also thank MKRS (Bumi) Sdn. Bhd. for all technical support.

## Author contributions

**Conceptualization:** Pei Pei Heng, Hanizah Mohd Yusoff.

**Data curation:** Pei Pei Heng.

**Formal analysis:** Pei Pei Heng.

**Funding acquisition:** Hanizah Mohd Yusoff.

**Investigation:** Pei Pei Heng.

**Methodology:** Pei Pei Heng, Hanizah Mohd Yusoff.

**Project administration:** Pei Pei Heng, Hanizah Mohd Yusoff, Mohamad Ridza Hj Illias.

**Resources:** Pei Pei Heng.

**Software:** Pei Pei Heng.

**Supervision:** Rozita Hod, Hanizah Mohd Yusoff.

**Validation:** Pei Pei Heng.

**Writing – original draft:** Pei Pei Heng.

**Writing – review & editing:** Hanizah Mohd Yusoff, Yap Jun Fai, Muhammad Fadhli Mohd Yusoff, Norizzati binti Amsah.

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
