## [Decision Letter · Decision Letter 0]

24 Jun 2025

Dear Dr. Heng,

Thank you for submitting your manuscript to PLOS ONE. After careful consideration, we feel that it has merit but does not fully meet PLOS ONE’s publication criteria as it currently stands. Therefore, we invite you to submit a revised version of the manuscript that addresses the points raised during the review process.

**ACADEMIC EDITOR:**

We look forward to receiving your revised manuscript.

Kind regards,

Emiliano Cè, Ph.D.

Academic Editor

PLOS ONE

Journal Requirements:

3. Thank you for stating the following financial disclosure: [This research was funded by the Universiti Kebangsaan Malaysia]. 

5. We notice that your supplementary file are included in the manuscript file. Please remove them and upload them with the file type 'Supporting Information'. Please ensure that each Supporting Information file has a legend listed in the manuscript after the references list.

6. Please include captions for your Supporting Information files at the end of your manuscript, and update any in-text citations to match accordingly. Please see our Supporting Information guidelines for more information: http://journals.plos.org/plosone/s/supporting-information .

Reviewers' comments:

Reviewer's Responses to Questions

**Comments to the Author**

1. Is the manuscript technically sound, and do the data support the conclusions?

Reviewer #1: Partly

Reviewer #2: Yes

2. Has the statistical analysis been performed appropriately and rigorously?

Reviewer #1: I Don't Know

Reviewer #2: Yes

3. Have the authors made all data underlying the findings in their manuscript fully available?

Reviewer #1: Yes

Reviewer #2: Yes

4. Is the manuscript presented in an intelligible fashion and written in standard English?

Reviewer #1: No

Reviewer #2: Yes

Reviewer #1: The findings are novel and potentially impactful for developing fatigue-based safety interventions. However, the manuscript requires revisions to enhance clarity, methodological transparency, and practical applicability;

1. The manuscript states that fatigue was measured "pre- and post-exposure to scaffolding task" on day 5, with an additional measurement "just before scaffolders participated in the next task on day 6," when unsafe behavior was assessed. However, it is unclear which fatigue measurements (post-day 5 or pre-day 6) were used in the correlation and regression analyses with unsafe behavior. This ambiguity affects the interpretation of results.

2. The unsafe behavior checklist, based on 38 standard protocols, is referenced as validated in a prior study (Yusoff et al., 2024). While this is noted, the manuscript lacks details on its reliability (e.g., inter-rater reliability) and validity (e.g., content or construct validity), which are critical for readers to assess its robustness.

Recommendation: Briefly summarize the checklist’s psychometric properties (e.g., "The checklist demonstrated high inter-rater reliability, kappa = 0.85, as reported in Yusoff et al., 2024") in the Methods section (lines 186-188) or refer readers to specific validation metrics in the cited study.

3. The Conclusion (lines 475-483) mentions that the test protocol can be used on-site to evaluate fatigue and predict safety performance, but the manuscript lacks specific guidance on implementation. For example, how feasible is it to conduct seven tests daily on a worksite? Which measures are most practical?

4. The study used a simulated task, which may not fully reflect real-world scaffolding conditions (e.g., variable weather, longer shifts). Additionally, the sample includes mostly inexperienced scaffolders (66.3% with <5 years’ experience, Table 2), potentially limiting applicability to seasoned workers.

5. Add recent literture 2024-25

6. The manuscript contains grammatical errors and awkward phrasing:

- Abstract (lines 37-38): "The knowledge gap on the relationship... is important in the development of accident prevention protocol" should be "Understanding the relationship... is crucial for developing accident prevention protocols."

- Introduction (lines 79-80): "Scaffold is a temporary structure..." should be "Scaffolds are temporary structures..."

Reviewer #2: Congratulations is a very nice article. Indeed.

I have no problem to endorse it for publications.

However, speaking about construction working place: the lack of education and the very bad lifestyle of poor people should not even be among the predominant/concern factors for accident.

In fact,

Scaffold has to be build by specialized workers = people that have been trained many times to do it.

Scaffold has to be validated by inspectors (third-party), prior the opening of the building site to the other workers.

The main problem that I see in your Country is the lack of a clear certification about proper safety operations in building site.

Moreover,

due to the nature of the tasks performed by the workers, they can't show up at the working site in bad physical and mental shape.

Inspectors of the working site should monitoring the status of the workers at the starting of their working shift.

**Do you want your identity to be public for this peer review?** For information about this choice, including consent withdrawal, please see our Privacy Policy

Reviewer #1: No

Reviewer #2: **Yes: ** DOMENICO FUOCO

---

## [Author Response · Author response to Decision Letter 1]

31 Oct 2025

PONE-D-25-24315: Predicting Unsafe behaviour from the Objective Assessment of Fatigue Manifestation among Scaffolders: Evidence from a Quasi-Experimental Simulation Study

Editor’s comments

Respond: The manuscript has been formatted to meet PLOS ONE's style requirements. The file naming has also been made on the title of the Ms Word document as ‘Predicting Unsafe behaviour from the Objective Assessment of Fatigue Manifestation among Scaffolders: Evidence from a Quasi-Experimental Simulation Study’.

Respond: We apologize for the error. The grant information has been standardized to ‘This research was funded by the Universiti Kebangsaan Malaysia’ in both sections.

3. Thank you for stating the following financial disclosure: [This research was funded by the Universiti Kebangsaan Malaysia]. Please state what role the funders took in the study. If the funders had no role, please state: ""The funders had no role in study design, data collection and analysis, decision to publish, or preparation of the manuscript."" If this statement is not correct you must amend it as needed.Please include this amended Role of Funder statement in your cover letter; we will change the online submission form on your behalf.

Respond: The following sentence has been newly added in the ‘Funding Statement’: The funders had no role in study design, data collection and analysis, decision to publish, or preparation of the manuscript. The amended Role of Funder statement also has been newly added in the cover letter.

4. We note that your Data Availability Statement is currently as follows: [All relevant data are within the manuscript and its Supporting Information files.]. Please confirm at this time whether or not your submission contains all raw data required to replicate the results of your study. Authors must share the “minimal data set” for their submission. PLOS defines the minimal data set to consist of the data required to replicate all study findings reported in the article, as well as related metadata and methods (https://journals.plos.org/plosone/s/data-availability#loc-minimal-data-set-definition).

Respond: Data cannot be shared publicly because of ethical (participants confidentiality) restriction. Even though the data have been anonymized, in combination they are potentially re-identifiable. Researchers of this study have to comply with the requirements of the Universiti Kebangsaan Malaysia Research Ethics Committee. Therefore, data are available upon reasonable request to the Universiti Kebangsaan Malaysia Research Ethics Committee (contact via sepukm@ukm.edu.my) for researchers who meet the criteria to access the confidential data. We have revised in the “Data availability statement”.

5. We notice that your supplementary file are included in the manuscript file. Please remove them and upload them with the file type 'Supporting Information'. Please ensure that each Supporting Information file has a legend listed in the manuscript after the references list.

Respond: We apologize for the mistake. The supplementary file has been uploaded separately as ‘Supporting Information’ with a legend listed in the manuscript after the reference list:

S1 Table. Validated Checklist for Individual Evaluation of Unsafe Behavior during the 2-Lifts Basic Static Scaffold Tower Erection Protocol

6. Please include captions for your Supporting Information files at the end of your manuscript, and update any in-text citations to match accordingly.

Respond: Captions for Supporting Information files has been included at the end of your manuscript, with the relevant in-text citation added to “Study instruments and measures” section under methodology: The complete checklist is available in the supplementary information file.

Respond: Reference list has been checked thoroughly to ensure completeness.

Reviewer 1’s comments

1. The findings are novel and potentially impactful for developing fatigue-based safety interventions. However, the manuscript requires revisions to enhance clarity, methodological transparency, and practical applicability:

The manuscript states that fatigue was measured "pre- and post-exposure to scaffolding task" on day 5, with an additional measurement "just before scaffolders participated in the next task on day 6," when unsafe behavior was assessed. However, it is unclear which fatigue measurements (post-day 5 or pre-day 6) were used in the correlation and regression analyses with unsafe behavior. This ambiguity affects the interpretation of results

Respond: Thank you for the comment.

We apologize for the confusion raised in our in-text description for the study protocol. The manuscript under “study protocol” section in the Methos, has been revised to ensure that this is clearly and consistently stated throughout.

We have also applied Figure 1 to illustrate the study flow:

From day1 to day 4, there was no hands-on exposure to any scaffolding task.

Day 5 was therefore considered as the baseline state and the exposure to scaffolding task was introduced from 9.00am- 6.00pm. This exposure on day 5, was hypothesized to generate fatigue. The fatigue manifestation following scaffolding task exposure was affirmed by the significant deterioration of fatigue parameters which were evaluated at pre (early morning day 5) and post-exposure (early morning of day 6).

This pre-post fatigue evaluation was not explained in details in this manuscript because it is not included as part of the manuscript’s objective.

However, our previous work on this pre-post finding has been published earlier in Pei et al. 2025 (new reference has been cited in this manuscript to acknowledge the readers).

Given the evidence that fatigue was manifested after a full day exposure on day 5, day 6 was therefore considered as the post exposure state (participants were in fatigue state). On day 6, following an early morning assessment of fatigue parameters, participants were asked to perform the simulated scaffolding task where the unsafe behavior was evaluated throughout the task. The evaluation of both independent and independent variables on similar day allowed the examination of correlation and regression.

We have revised the in-text explanation under study protocol:

“Practical session was initiated on day 5 which indicated an exposure phase, where participants were exposed to the repeated hands-on cycle of scaffold erection-dismantling procedures from 9.00 am to 6.00pm, with 3 standardized breaks in between. Hence, this exposure on day 5 was hypothesized to generate fatigue. A 14-hours of overnight rest were allowed until they came back to training site the next day (day 6). Fatigue generation during work exposure on day 5, and the carry-over effects of fatigue from one work day (day 5) to the next before scaffolders are allowed to initiate next task (day 6), have been affirmed. Our evidence on pre-post deterioration in fatigue parameters during this study phase had been published earlier by Pei et al. 2025.In the post-exposure state where fatigue had developed (day 6), fatigue manifestation was measured early in the morning, followed by their participation in simulated scaffold erection task from 9.00am-12.00pm.The impaired safety performance in term of unsafe behavior was observed and evaluated at individual level throughout this simulated work”.

(Line 147-162)

Reference:

Pei, H.P., Mohd Yusoff, H., Hj Illias, M.R., Karrupayah, S., Mohd Yusoff, M.F. and Hod, R., 2025. Work-related fatigue among scaffolders as indicated by physical and cognitive tasks: objective fatigue assessment from a single group experimental study. Fatigue: Biomedicine, Health & Behavior, 13(2), pp.160-171.

2. The unsafe behavior checklist, based on 38 standard protocols, is referenced as validated in a prior study (Yusoff et al., 2024). While this is noted, the manuscript lacks details on its reliability (e.g., inter-rater reliability) and validity (e.g., content or construct validity), which are critical for readers to assess its robustness.

Recommendation: Briefly summarize the checklist’s psychometric properties (e.g., "The checklist demonstrated high inter-rater reliability, kappa = 0.85, as reported in Yusoff et al., 2024") in the Methods section (lines 186-188) or refer readers to specific validation metrics in the cited study.

Respond: Thank you for the comment.

Different from majority of the construct-based instruments which are used to measure the unobservable psychological traits or behavioral characteristics, our unsafe behavioral checklist is the procedural tool which direct measures the observable actions. The focus is on the process, but not the theoretical reason behind it.

The validity and acceptability of procedural checklist is best performed using Fuzzy Delphi Method (FDM) analysis rated by panel of experts within the study scope, as what have been reported in our prior study of Yusoff et al. 2024.

The procedural tool is considered valid and acceptable when FDM analysis adheres to three main prerequisites; firstly, the experts’ consensus for each construct is fulfilled by a threshold value (d) ≤ 0.2. Secondly, the experts’ consensus for each item is fulfilled at 75 %, while the third prerequisite aims to rank the items using average fuzzy numbers where items with lower ranks need to be discarded.

We have summarized the checklist’s acceptability metrics in the Method section as recommended:

“The procedural checklist demonstrated high experts' consensus for each construct with threshold value (d) ≤ 0.2; and for all items with agreement of ≥75 %. The average fuzzy numbers documented ranging from 0.588 to 0.8. None of the items with the lowest ranking was discarded as all items perfectly fulfilled the second prerequisite and obtained excellent experts’ agreement.”

(Line 204-213)

Reference:

Jailani, M.A. and Loy, C.K., 2023. The Application of Fuzzy Delphi Method in Content Validity Analysis. International Association for Development of the Information Society.

Jung, C.F., Breaud, A.H., Sheng, A.Y., Byrne, M.W., Muruganandan, K.M., Dhanani, M. and Leo, M.M., 2016. Delphi method validation of a procedural performance checklist for insertion of an ultrasound-guided peripheral intravenous catheter. The American Journal of Emergency Medicine, 34(11), pp.2227-2230.

Yusoff, H.M., Heng, P.P., Illias, M.R.H., Karrupayah, S., Fadhli, M.A. and Hod, R., 2024. A qualitative exploration and a Fuzzy Delphi validation of high-risk scaffolding tasks and fatigue-related safety behavioural deviation among scaffolders. Heliyon, 10(15).

3. The Conclusion (lines 475-483) mentions that the test protocol can be used on-site to evaluate fatigue and predict safety performance, but the manuscript lacks specific guidance on implementation. For example, how feasible is it to conduct seven tests daily on a worksite? Which measures are most practical?

Respond: Thank you for the comment.

In line with our study objective which primarily aims to determine the relationship between fatigue manifestation and unsafe behavior among scaffolders, we evidently reported the prediction model generated from regression analysis which has confirmed the four (4) significant fatigue predictors of unsafe behaviour:

1. Simple reaction time

2. One leg standing duration with eyes closed

3. Trunk flexor endurance duration

4. PSQI sleep quality screening

This indicates that out of the seven (7) objective tests employed based on their practicality on-site as shown in literatures, four (4) were demonstrated significant association with rate of unsafe behaviour. Hence, we did clarify this in the conclusion by stating that those four tests could be practically employed on-site for safety performance prediction.

Nevertheless, this was a preliminary study exploring this association experimentally. Therefore, additional scope like the protocol implementation on site, and the feasibility of conduction is not part of the study objective therefore are not being covered in present manuscript. However, thank you for the kind recommendation and we have included this in the conclusion section as recommendations for future study, as an extension to present study:

“The quasi-experiment had refined the significant relationship between physical and cognitive fatigue manifestation, and the rate of unsafe behaviour. The prediction model had confirmed four significant fatigue predictors of unsafe behaviour, including simple reaction time, one leg standing duration with eyes closed, trunk flexor endurance duration and PSQI sleep quality screening, indicating these testscan be practically used as the on-site evaluation of fatigue and anticipate safety performance before scaffolding task is assigned to scaffolder. Thestudy findings will assist industry managers and regulatory decision-makers to put in place appropriate measures to govern workers' safe behaviour, besides guiding the formulation of fatigue intervention and training among scaffolders, which eventually helps in accidents’ prevention. As an extension to present study, future studies are needed to explore the tests’ application and feasibility of conduction in the real scaffolding setting”.

(Line 558-569)

4. The study used a simulated task, which may not fully reflect real-world scaffolding conditions (e.g., variable weather, longer shifts). Additionally, the sample includes mostly inexperienced scaffolders (66.3% with <5 years’ experience, Table 2), potentially limiting applicability to seasoned workers.

Respond: Thank you for the comment.

In present study, external validity and generalizability were maintained to the best of our limit, as there was adequate examination of work characteristics of scaffolding workers.

The experimental setting/ study site was also mimics the real scaffolding site therefore, it could be possible to interpret findings among other groups of scaffolding workforces. The outcome of study could be specifically examined by the level of fidelity when the experiment was conducted as intended.

Nevertheless, we admit there were several limitations. The inference from our sample to the target population of scaffolders might be slightly inadequate, because present study recruited participants who attended scaffolding training on the training site in order to ease the conduction and control of the quasi-experiment. Although the work characteristics are similar between study population and the target population, there might be variations in term of individual sociodemographic and job characteristics such as age group, comorbidity, work experience and literacy level. Our study somehow was a preliminary exploration on this research topic. Based on the positive outputs generated and realizing all limitations of study, we therefore recommend future studies to be expanded in the real worksite.

Various environmental factors including noise, lighting, vibration, extreme temperature and working at height have been reported associated with work fatigue. However, the environment factors were controlled in the quasi-experiment in such a way that all participants performed simulation task on the same training site hence the climatic factors were equally applied to all.

Thank you for raising a concern on this. We have added a brief discussion on this limitation in the text:

“This is the first study employed objective measurement tool to assess fatigue manifestation, and refined its relationship with uns

---

## [Decision Letter · Decision Letter 1]

1 Dec 2025

Predicting Unsafe behaviour from the Objective Assessment of Fatigue Manifestation among Scaffolders: Evidence from a Quasi-Experimental Simulation Study

PONE-D-25-24315R1

Dear Dr. Heng,

We’re pleased to inform you that your manuscript has been judged scientifically suitable for publication and will be formally accepted for publication once it meets all outstanding technical requirements.

Kind regards,

Emiliano Cè, Ph.D.

Academic Editor

PLOS ONE

Additional Editor Comments (optional):

Reviewers' comments:

Reviewer's Responses to Questions

**Comments to the Author**

Reviewer #3: All comments have been addressed

2. Is the manuscript technically sound, and do the data support the conclusions?

Reviewer #3: Yes

3. Has the statistical analysis been performed appropriately and rigorously?

Reviewer #3: Yes

4. Have the authors made all data underlying the findings in their manuscript fully available?

Reviewer #3: Yes

5. Is the manuscript presented in an intelligible fashion and written in standard English?

Reviewer #3: Yes

Reviewer #3: All the questions have been addressed. I have no further concerns. No dual publication, research ethics, or publication ethics issue.

**Do you want your identity to be public for this peer review?** For information about this choice, including consent withdrawal, please see our Privacy Policy

Reviewer #3: **Yes: ** Wenfei Zhu

---

## [Editor Report · Acceptance letter]

PONE-D-25-24315R1

PLOS One

Dear Dr. Heng,

I'm pleased to inform you that your manuscript has been deemed suitable for publication in PLOS One. Congratulations! Your manuscript is now being handed over to our production team.

Kind regards,

on behalf of

Prof. Emiliano Cè

Academic Editor

PLOS One